# Survey of Piroplasmids in Wild Mammals, Unconventional Pets, and Ticks from Goiás State, Midwestern Brazil

**DOI:** 10.3390/pathogens14060585

**Published:** 2025-06-12

**Authors:** Raphaela Bueno Mendes Bittencourt, Ana Cláudia Calchi, Lucianne Cardoso Neves, Nicolas Jalowitzki de Lima, Gabriel Cândido dos Santos, Ennya Rafaella Neves Cardoso, Warley Vieira de Freitas Paula, Luciana Batalha de Miranda Araújo, Jessica Rocha Gonçalves, Elisângela de Albuquerque Sobreira, Luiz Alfredo Martins Lopes Baptista, Hermes Ribeiro Luz, Marcos Rogério André, Filipe Dantas-Torres, Felipe da Silva Krawczak

**Affiliations:** 1Veterinary and Animal Science School, Federal University of Goiás, Goiânia 74605-220, GO, Brazil; rafabmbitt@discente.ufg.br (R.B.M.B.); luciannecardoso@egresso.ufg.br (L.C.N.); jalowitzki@discente.ufg.br (N.J.d.L.); doscandido@discente.ufg.br (G.C.d.S.); ennyaneves@discente.ufg.br (E.R.N.C.); warleyvieira@egresso.ufg.br (W.V.d.F.P.); batalha@ufg.br (L.B.d.M.A.); rochajessica@discente.ufg.br (J.R.G.); 2Vector-Borne Bioagents Laboratory (VBBL), Department of Pathology, Reproduction and One Health, Faculty of Agricultural and Veterinary Sciences, São Paulo State University (UNESP), Jaboticabal 14884-900, SP, Brazil; ana.calchi@unesp.br (A.C.C.); mr.andre@unesp.br (M.R.A.); 3Environmental Secretariat of the Municipality of Anápolis (SMA), Anápolis 75113-180, GO, Brazil; elisangela@anapolis.go.gov.br; 4Brazilian Institute of Environment and Renewable Natural Resources (IBAMA), Wild Animal Screening Center of Goiás (CETAS-GO), Goiânia 74605-090, GO, Brazil; luiz.baptista@ibama.gov.br; 5Post-Graduation Program in Health and Environment, Biodiversity and Conservation and Northeast Biotechnology Network (RenorBio), Federal University of Maranhão, São Luís 65085-580, MA, Brazil; hermes.luz@ufma.br; 6Laboratory of Immunoparasitology, Department of Immunology, Aggeu Magalhães Institute, Oswaldo Cruz Foundation (Fiocruz), Recife 50740-465, PE, Brazil; fdtvet@gmail.com

**Keywords:** tick-borne infectious agents, *Cytauxzoon brasiliensis*, *Theileria* spp., *Babesia goianiaensis*, Brazil

## Abstract

Tick-borne piroplasmids are apicomplexan protozoa that infect a wide range of vertebrate hosts, with significant implications for animal and human health. This study investigated the occurrence and genetic diversity of piroplasmids in wild mammals, unconventional pets, and associated ticks in Goiás state, midwestern Brazil. Between April 2023 and January 2024, 105 blood samples, 22 tissue samples, and 300 ticks were collected from 21 mammalian species housed in wildlife screening centers, zoos, and veterinary clinics. Molecular screening targeting the 18S rRNA gene of piroplasmids detected a 25.7% (27/105) overall positivity, with gray brockets (*Subulo gouazoubira*) and South American tapirs (*Tapirus terrestris*) showing the highest infection rates. Three tick samples tested positive, including two *Amblyomma sculptum* nymphs and a male of *Amblyomma dubitatum* collected from a tapir and capybara (*Hydrochoerus hydrochaeris*)*. Cytauxzoon brasiliensis* was reported, for the first time, in cougars (*Puma concolor*) from Goiás state, midwestern Brazil, indicating the role of this feline as a host of this parasite. *Babesia goianiaensis* was confirmed in a capybara, and *Theileria terrestris* in tapirs. Phylogenetic analyses clustered gray brockets-associated *Theileria* sequences with *Theileria* sp. previously detected in Neotropical deer from Brazil and *Theileria cervi*. While the phylogenetic analysis of amino acid sequences of the cytochrome c oxidase subunit III separated *Theileria* genotypes detected in *S. gouazoubira* from *T. cervi*, *hsp70*-based phylogenetic inferences clustered the genotypes detected in *Tapirus terrestris* with *Theileria terrestris*, suggesting host-specific evolutionary lineages. These findings contribute to the understanding of Piroplasmida diversity and circulation in South American wild mammals, emphasizing the need for enhanced molecular surveillance to elucidate transmission dynamics, assess potential health risks, and contribute to the establishment of wildlife conservation and One Health strategies.

## 1. Introduction

Tick-borne piroplasmids are apicomplexan protozoa that infect wild mammals worldwide, including captive, synanthropic, and free-living animals. Most available reports indicate an absence of clinical manifestations of piroplasmosis in free-living and captive animals [1,2,3].

The order Piroplasmida comprises apicomplexan protozoan parasites belonging to the genera *Babesia*, *Cytauxzoon*, and *Theileria*, which are amongst the most widespread hemoparasites globally and infect a diverse range of hosts, including many mammal species [4,5]. Several piroplasmids have been identified in domestic and wild animals, with some detected in humans [3,5,6]. For instance, babesiosis and theileriosis significantly impact companion animals, livestock, and free-living wildlife, with some species presenting zoonotic potential [4,6,7]. While ixodid ticks are the primary vectors for piroplasmids through blood feeding, wild mammals are key hosts for both ticks and piroplasmids [5,8,9].

Despite the importance of various wild mammals as hosts of piroplasmids, their role in the transmission cycles of these agents remains poorly understood. This gap in knowledge underscores the need for further research to unravel the diversity of piroplasmids in wildlife [3,5,10,11]. Detecting diverse piroplasmids in Brazilian wild mammals may reveal the intricate interactions among protozoa, tick vectors, and wildlife as well as the potential risk of spillover to domestic animals and humans [2,5,11,12,13,14]. Monitoring wildlife populations and understanding parasite-host relationships are crucial for wild mammal conservation and public health strategies [10,11,15].

Therefore, this study aimed to investigate the occurrence and diversity of piroplasmids in wildlife and unconventional mammal pets from wildlife rescue centers and veterinary clinics in Goiás state, midwestern Brazil.

## 2. Materials and Methods

### 2.1. Ethical Aspects

The study was approved by the Institutional Animal Care and Use Committee (CEUA/UFG) of the Federal University of Goiás (protocol 122/22). As it involved only captive animals, the Chico Mendes Institute for Biodiversity (ICMBio) exempted it from authorization through the Biodiversity Authorization and Information System.

### 2.2. Study Area

This study was conducted in the state of Goiás, located in midwestern Brazil. This region is characterized by a tropical climate in the Cerrado biome, a vast tropical savanna ecoregion recognized as a global biodiversity hotspot [16].

Sampling was performed in the Wild Animal Screening Center of Goiás (CETAS-GO), located in the municipality of Goiânia, the Screening and Rehabilitation Center for Wild Animals of the municipality of Caldas Novas (CETRAS-CN), the Environmental Secretariat of the municipality of Anápolis (SMA), Goiânia Zoo (ZOO GYN), and veterinary clinics specialized in unconventional pets in Goiânia. These locations receive animals from across the state, mainly through confiscation, rescue operations, or clinical care. The origin of each animal was determined based on information provided in admission forms, including the capture location.

Animals originated from 30 municipalities within Goiás (Figure 1), with the highest numbers from Goiânia (46), Caldas Novas (9), Anápolis (4), and Bela Vista de Goiás (3). Additionally, animals were received from two municipalities bordering Goiás: Cassilândia (Mato Grosso) and Paranaíba (Mato Grosso do Sul), both located in midwestern Brazil (Figure 1).

### 2.3. Sampling

Blood, tissue, and ectoparasite sampling was conducted at the designated screening centers (CETAS-GO, CETRAS-CN, SMA, ZOO GYN) and veterinary clinics. Between April 2023 and January 2024, 105 blood samples and 22 tissue samples were collected. Sampled animals included free-living wildlife (*n* = 94) from rehabilitation and screening centers, zoo animals (*n* = 5), and unconventional pets (*n* = 6) (Table 1). All samples were transported to the Laboratory of Parasitic Diseases (LADOPAR) at the School of Veterinary and Animal Science, Federal University of Goiás (EVZ/UFG).

Blood samples were collected in EDTA tubes, frozen, and stored at −20 °C until molecular analysis. Tissue samples were collected during necropsies of ten animals from CETAS-GO, including one cougar *Puma concolor* (lung, liver, and spleen), two Brazilian porcupines *Coendou prehensilis* (spleen), one South American tapir *Tapirus terrestris* (lung, liver, and spleen), one white-eared opossum *Didelphis albiventris* (liver and spleen), two capybaras *Hydrochoerus hydrochaeris* (lung, liver, and spleen), one crab-eating fox *Cerdocyon thous* (lung and liver), and two maned wolves *Chrysocyon brachyurus* (lung, liver, and spleen). These tissue samples were frozen and stored at −20 °C until molecular analysis. Additionally, blood samples from all these animals were included in the study.

### 2.4. Microscopic Analysis of Blood Smear Samples

Of the 105 samples, 30.5% (32/105) arrived at the laboratory fresh and refrigerated. Duplicate blood smears were immediately prepared following blood collection and stained using a rapid staining kit (Panótico Rápido, Laborclin, Pinhais, PR, Brazil). The smears were then examined cytologically under a standard light microscope (Olympus BX41, Olympus Optical Co., Ltd., Tokyo, Japan) at 1000× magnification and photographed using a digital camera.

### 2.5. Tick Collection

A total of 300 ticks were collected from individual hosts, spanning nine animal species (13 *Myrmecophaga tridactyla*, four *Tapirus terrestris*, four *P. concolor*, five *H. hydrochaeris*, three *C. prehensilis*, four *Tamandua tetradactyla*, one *C. thous*, one *Alouatta caraya*, and two *Subulo gouazoubira*). Out of 300 ticks, 186 were collected from 25 animals whose blood samples were also obtained, whereas 114 were collected from 12 wild mammals whose blood samples could not be collected due to logistical constraints related to collection sites or animals.

All ticks collected from animals were identified using previously described taxonomic keys. While nymphs and adults were identified at the species level [17,18], larvae were identified at the genus level [19]. In total, 80 ticks (65 *Amblyomma sculptum*, seven *Amblyomma dubitatum*, eight *Amblyomma nodosum*), including nymphs and adults, were individually processed for DNA extraction and PCR analysis. The remaining specimens were deposited in the “Coleção Nacional de Carrapatos do Cerrado” (CNCC) Marcelo Bahia Labruna of the Veterinary and Animal Science School, Federal University of Goiás.

### 2.6. DNA Extraction

DNA was extracted from blood and tissue samples using Blood Genomic Prep Mini Spin Kit (Cytiva, Marlow, UK) and DNeasy Blood & Tissue Kit (Qiagen, Valencia, Santa Clarita, CA, USA), respectively, according to the manufacturer’s recommendations. DNA from 80 ticks was extracted using the guanidine isothiocyanate-phenol-chloroform protocol [20]. Each DNA extraction batch included microtubes with distilled water as a negative control. DNA samples were placed into sterile 1.5 mL DNAse/RNAse-free polypropylene tubes, identified, and stored at –20 °C until PCR analysis.

### 2.7. Molecular Analyses

Blood, tissue, and tick DNA samples were subjected to PCR assays targeting the mammalian *cytB* gene [21] and tick 16S rRNA gene [22] as endogenous controls to rule out the occurrence of false negative results.

All biological samples (blood, tissue, and ticks) were initially screened by a PCR assay targeting a fragment of 551 bp of the 18S rRNA gene of piroplasmids [23]. Feline biological samples that tested positive in the abovementioned PCR were also subjected to a PCR for *Cytauxzoon* spp. based on a 284 bp fragment of the 18S rRNA gene [23]. PCR products were subjected to electrophoresis on a 1.5% agarose gel (Sinapse Inc., Hollywood, FL, USA). Following the manufacturer’s instructions, the gels were stained with SYBR Safe^®^ (Invitrogen, Carlsbad, CA, USA). Electrophoresis was performed in a horizontal chamber using 0.5X TBE buffer (0.045 M Tris-borate; 0.001 M EDTA, pH 8.0) at 1 to 10 V/cm for 90 min. Gel visualization was conducted using an transilluminator (Kasvi^®^, São José dos Pinhais, Brazil).

Positive samples were subjected to additional PCR testing targeting the following genetic markers for piroplasmids: near-complete 18S rRNA (~1500 bp) [24,25], *cox-1* (~1000 bp) [26], *cox-3* (~700 bp) [26,27]; *hsp70* (~900 bp) [28] and intergenic spacer (ITS-1) (~450 bp) [29]; and for of *Cytauxzoon* spp.: *cox-1* (1656 bp) [30] and *cytB* (1434 bp) [31]. PCR products were visualized by electrophoresis as previously described.

### 2.8. Amplicon Purification, Sequencing, and Phylogenetic Analyses

The short 18S rRNA amplicons were purified using the Wizard^®^ SV Gel and PCR Clean-Up System (Promega, Madison, WI, USA) and bidirectionally sequenced using the BigDye TM Terminator v3.1 Matrix Standards Kit^®^ (Applied Biosystems, Foster City, CA, USA). Sequencing reactions were performed at the Aggeu Magalhães Institute, Oswaldo Cruz Foundation (Fiocruz Pernambuco) using a 3500× L Genetic Analyzer (Applied Biosystems, Foster City, CA, USA), employing the same primers utilized in the PCR assays. Sequences were trimmed (error probability = 0.05) and assembled using Clustal Omega in Geneious Prime version 2025.0.3 (https://www.geneious.com, accessed on 26 February 2025). Sequence similarity searching was performed using the Basic Local Alignment Search Tool (BLASTn version 2.16.0) (https://blast.ncbi.nlm.nih.gov/Blast.cgi, accessed on 26 February 2025) to assess their identity with sequences in the GenBank database.

Long 18S rRNA amplicons and other amplified products displaying high-intensity bands in agarose gel electrophoresis were purified using the ExoSAP-IT PCR Product Cleanup Reagent^®^ (Applied Biosystems, Foster City, CA, USA) and sequenced. Sequencing was performed using the dideoxynucleotide chain termination method at the Human Genome and Stem Cell Research Center, University of São Paulo (USP), São Paulo, Brazil. The obtained sequences were subjected to a quality screening process using Phred-Phrap software (version 23) to evaluate electropherogram quality and generate consensus sequences from the alignment of sense and antisense strands [32]. Sequence identity was assessed using BLASTn by comparing the obtained sequences with those deposited in GenBank. Sequences saved in FASTA format were aligned with homologous sequences retrieved from GenBank using MAFFT and edited with BioEdit v.7.0.5.3 [33]. For phylogenetic inference of *cox-3* gene sequences, a 242-amino-acid alignment was used in combination with the mtZOA+G evolutionary model. For phylogenetic inference, the following alignments and evolutionary models were used: a 1610 bp alignment of the 18S rRNA gene (TIM3+I+G model), a 909 bp alignment of the *hsp70* gene (TIM3+I+G model), a 242-amino-acid alignment of the *cox-3* gene (mtZOA+G model), and a 2851-character alignment of the ITS-1 region (HKY+I+G model). These models were selected based on the Bayesian Information Criterion (BIC) using the W-IQ-Tree web server (available online: http://iqtree.cibiv.univie.ac.at/ accessed on 26 February 2025) [34], and phylogenetic analyses were conducted using the maximum likelihood method with 1000 bootstrap replicates. The phylogenetic trees were edited using TreeGraph 2.0.56-381 beta software [35]. The topology and identification of the phylogenetic clades were based on previous studies [11,12,14,36].

## 3. Results

### 3.1. Blood Smear Analysis

Among the 36 stained-blood smears analyzed, only one from a cougar exhibited intracytoplasmic inclusions consistent with *Cytauxzoon* spp. (Figure 2). Subsequent molecular analysis by PCR confirmed the presence of *Cytauxzoon* spp. DNA in this sample.

### 3.2. Tick Identification

Ticks (*n* = 300) belonged to nine species and were collected from 37 mammals of nine species. The most prevalent tick species was *A. sculptum*, comprising 65.4% (196/300) of the collected ticks, including 123 nymphs, 40 males, and 33 females. Other identified species included *Amblyomma dubitatum* (12.7%; 38/300), *Amblyomma nodosum* (8.3%; 25/300), *Amblyomma ovale* (1.3%; 4/300), *Amblyomma calcaratum* (1.3%; 4/300), *Amblyomma longirostre* (1.3%; 4/300), *Rhipicephalus microplus* (1.0%; 3/300), and *Dermacentor nitens* (0.7%; 2/300). Additionally, 8.0% (24/300) of the specimens were identified as *Amblyomma* spp. larvae (Table 2).

The following voucher tick specimens were deposited in the tick collection “Coleção Nacional de Carrapatos do Cerrado” (CNCC) Marcelo Bahia Labruna of the Veterinary and Animal Science School, Federal University of Goiás (CNCC 104-120): six *Amblyomma* spp. larvae (CNCC 111 and CNCC 114), ten *A. dubitatum* adults (CNCC 104, CNCC 106-107), 43 *A. sculptum* nymphs (CNCC 107-109, CNCC 111, CNCC 114, and CNCC 117-120), 46 *A. sculptum* adults (CNCC 105, CNCC 109, CNCC 111, and CNCC 114), three *R. microplus* adults (CNCC 108 and CNCC 113), four *A. calcaratum* adults (CNCC 112), 21 *A. nodosum* adults (CNCC 110 and CNCC 118), four *A. ovale* adults (CNCC 115), two *D. nitens* adults (CNCC 116), and four *A. longirostre* adults (CNCC 117).

### 3.3. Molecular Detection of Piroplasmids in Ticks, Blood, and Tissue Samples

All blood and tissue samples tested positive by cPCR targeting the cytB gene, confirming the integrity and quality of the DNA extraction procedures used in this study. Among tick DNA samples tested, 86.2% (69/80) were positive for the 16S rRNA gene (control endogenous); negative samples were excluded from subsequent analyses. The following tick samples were tested for the 18S rRNA gene of piroplasmids: 59 *A. sculptum* (15 males, seven females, and 37 nymphs), five adult males of *A. dubitatum*, and five adult males of *A. nodosum*. Among the 69 tested ticks, 4.3% (3/69) were positive (Table 2). Two *A. sculptum* nymphs collected from the same *Tapirus terrestris* tested positive, aligning with a positive PCR result from its blood sample. Additionally, one adult male *A. dubitatum* collected from *H. hydrochaeris* tested positive, although the blood sample from this host was negative.

Molecular analysis of blood samples detected piroplasmid DNA in 25.7% (27/105) of tested mammals, distributed across nine species: *C. thous* (*n* = 2), *C. brachyurus* (*n* = 2), *C. prehensilis* (*n* = 1), *H. hydrochaeris* (*n* = 2), *M. americana* (*n* = 1), *S. gouazoubira* (*n* = 9), *M. tridactyla* (*n* = 2), *P. concolor* (*n* = 4), and *Tapirus terrestris* (*n* = 4) (Table 3). Only *P. concolor* individuals tested positive for piroplasmids in PCR assays among the seven sampled felids.

Tissue samples from *Tapirus terrestris* (liver and spleen) and *P. concolor* (lung) were also positive for piroplasmid DNA.

### 3.4. Phylogenetic and BLASTn Analyses

Piroplasmida 18S rRNA (1270 to 1397 bp) amplified from the blood of four *S. gouazoubira*, two *Tapirus terrestris*, and one *P. concolor* were successfully sequenced (Table 4). Based on a 1610 bp alignment and using the TIM3+I+G evolutionary model, a maximum likelihood phylogenetic analysis positioned the sequences from gray brockets within the *Theileria* sensu stricto clade. These sequences were phylogenetically related to *Theileria* sp. detected in deer from Brazil and Argentina (*S. gouazoubira*, *Blastocerus dichotomus*, and *Mazama rufa*) and *Theileria cervi* detected in deer from the USA. One sequence formed a clade with a *Theileria* sp. sequence detected in the same deer species, but this clade was a sister clade to the *Theileria orientalis* clade. The sequences detected in *Tapirus terrestris* were positioned within the *Theileria terrestris* group. In contrast, the sequence detected in *P. concolor* was placed in the *Cytauxzoon* clade, phylogenetically closer to *Cytauxzoon brasiliensis* (Figure 3). All these clades were supported by bootstrap values above 80%. The BLASTn analyses corroborated the phylogenetic positioning, showing identities >99% with *T. cervi*, 100% with *Theileria terrestris*, and 99.76% with *C. brasiliensis* (Table 4).

Additional genetic markers further supported these findings. Three *hsp70* sequences were obtained (one from *S. gouazoubira* and two from *Tapirus terrestris*—739 to 893 bp). The deer-associated sequence showed 92.9% identity with *T. cervi* detected in *Odocoileus virginianus* from the USA. In comparison, the sequences detected in *Tapirus terrestris* had a 99.04% and 99.22% identity with *Theileria terrestris*, which had been previously detected in tapirs from Brazil (Table 4). Phylogenetic analysis using maximum likelihood (909 bp alignment and TIM3+I+G as evolutionary model) positioned the deer-associated *hsp70* sequence within the *Theileria* sensu stricto clade and close to *T. cervi*. In contrast, tapir-associated sequences were placed in the *Theileria terrestris* group. These clades were supported by bootstrap values above 83% (Figure 4).

In the maximum likelihood phylogenetic inference, based on a 242 amino acid alignment of the *cox3* gene and using the mtZOA+G evolutionary model, three *Theileria* sp. COX3 sequences from *S. gouazoubira* clustered closely with genotypes previously detected in *B. dichotomus* from Brazil, albeit split into two subclusters related to the host species, supported by bootstrap values of 98% and 99% (Figure 5). The BLASTn analysis showed 87.50% and 87.62% identities with sequences previously detected in *B. dichotomus* (Table 4).

The phylogenetic analysis based on the alignment of 2851 characters of the intergenic spacer (ITS-1), using the HKY+I+G evolutionary model, revealed that the *Theileria* sp. sequences detected in *S. gouazoubira* from Brazil formed a well-supported monophyletic clade. This clade clustered closely with sequences of *Theileria* sp. detected in *Amblyomma americanum* in the USA (Figure 6). The bootstrap support values reinforced the reliability of this clade. The BLASTn analysis showed 89.17% and 89.60% identity to *Theileria* sp. previously detected in *A. americanum* (Table 4).

All samples subjected to PCR assays based on the *cox-1* and *cytB* genes showed negative results, precluding phylogenetic inferences based on these two additional molecular markers.

Although 30 samples (27 blood and 3 ticks) tested positive for piroplasmid DNA, only a subset yielded high-quality sequences suitable for phylogenetic analyses. All positive samples were initially subjected to amplification and sequencing of multiple genetic markers. However, sequencing success was limited by low DNA concentration and/or poor amplicon quality in several cases, particularly for longer or more variable gene regions (e.g., near-complete 18S rRNA, *hsp70*, ITS-1, and mitochondrial markers). These technical limitations, commonly encountered in wildlife studies involving opportunistic sampling and field-preserved materials, restricted the sequencing rate and may have contributed to the underrepresentation of certain genotypes in the phylogenetic trees.

Short-fragment sequencing of the 18S rRNA gene provided additional insights. One sequence detected in an *H. hydrochaeris* sample exhibited 100% identity with *Babesia goianiaensis* (OR149999.1), while a sequence detected in an *S. gouazoubira* sample showed 100% identity with *T. cervi* from the USA (AY735119.1). A sequence detected in *C. prehensilis* exhibited 100% identity with *Babesia vogeli* (MN912677.1). Due to their short size, these sequences were not included in the previously conducted phylogenetic analyses.

## 4. Discussion

Our study expands the understanding of piroplasmids in wild mammals from Brazil. Overall, its findings reinforce the ecological complexity of tick-borne pathogens, the potential risk of spillover to domestic animals and humans, and the need for continued surveillance efforts [36,37,38]. An interesting example in the present study is the probable detection of *B. vogeli* in a Brazilian porcupine. This piroplasmid is typical of canids and is mainly transmitted by brown dog ticks (*Rhipicephalus linnaei*) [39]. Although brown dog ticks prefer dogs and wild canids, they have been accidentally found on wildlife, including margay (*Leopardus wiedii*) [40], coati (*Nasua nasua*) [41], black-tailed marmoset (*Mico melanurus*) [42], and Brazilian guinea pig (*Cavia aperea*) [43], but never on Brazilian porcupines. Nonetheless, considering the short fragment size (304 bp), further research is needed to confirm if the sequence detected herein corresponds to *B. vogeli* or a closely related species. None of the piroplasmid-positive animals exhibited clinical signs specifically attributable to infection at the time of sampling.

The overall prevalence of piroplasmids in mammals observed in this study (25.7%) is consistent with previous reports from captive wild mammals in the states of Minas Gerais and Goiás, Brazil, where a positivity rate of 26.3% was recorded in animals housed in rehabilitation centers [1]. However, parasite-host associations can vary significantly across different regions for several reasons, including ecological and climatic factors influencing vector distribution [44].

This study confirms previously reported tick-host associations in Brazilian wildlife. For instance, the association between *A. sculptum* and eight mammal species corroborates previous studies. Other notable findings include *R. microplus* on *S. gouazoubira* and *Tapirus terrestris*; *A. nodosum* on *M. tridactyla* and *T. tetradactyla*, and *D. nitens* on *P. concolor*. Additionally, *A. longirostre* on *C. prehensilis*, *A. ovale* on *P. concolor*, and *A. calcaratum* on *M. tridactyla* and *T. tetradactyla*, are known tick-host associations [2,40,41,45]. In addition, the detection of *Amblyomma* spp. larvae in multiple wild animal species reinforces these mammals’ role in maintaining tick populations. However, tick infestation rates were lower than those documented in free-ranging animals [13,40,45,46], likely due to ectoparasite management practices in captive settings. Our findings contribute to a broader understanding of tick-host relationships in Brazilian mammals in midwestern Brazil.

This study conducted sample collection opportunistically, with information on the animal’s origin obtained post-capture. As a result, the capture locations cannot be reliably used to determine definitive prevalence or distribution patterns. Many wild mammals have extensive home ranges that may extend far beyond the capture site [47].

The molecular approach used herein focused on amplifying a fragment of the 18S rRNA gene, a widely used molecular target due to its conservation across piroplasmid species [11,12,48]. This strategy provided valuable insights into the parasite-host relationship and phylogenetic reconstruction. Phylogenetic analyses confirmed the distinct clustering of piroplasmids according to their respective hosts and reinforced the genetic diversity among piroplasmid species infecting wild mammals in midwestern Brazil.

Among the positive deer, 77% (10/13) originated from nine *S. gouazoubira* and one *M. americana*. This high positivity finding is consistent with a previous study [36], in which a 75.1% (136/181) positivity rate for piroplasmids was reported in Neotropical deer from Brazil. Phylogenetic analysis positioned the two 18S rRNA sequences obtained from *S. gouazoubira* in distinct clades. One of the sequences was related to *Theileria* spp. genotypes detected in deer from Brazil and *T. cervi*, whereas the other clustered with a sequence detected in the same host species, forming a sister clade to *T. orientalis*. These findings are consistent with those recently reported by Calchi et al. [36], evidencing the circulation of several *Theileria* genotypes and/or species in deer from Brazil. Interestingly, the *hsp70* sequence detected herein was more closely related to *T. cervi* than those genotypes already detected in Neotropical deer in the country [36]. Notably, the *hsp70* sequences obtained in the present study were longer than those obtained in the reported study [36], which might have influenced the phylogenetic positioning. Additionally, the three COX-3 amino acid sequences detected in *S. gouazoubira* were positioned in the same clade as sequences previously detected in *B. dichotomus*, albeit positioned apart in the subclade and showing a clear separation between the genotypes according to the host species (Figure 5). These findings indicate a possible association between *Theileria* genotypes and deer host species [36].

The known vector of *T. cervi* is the lone star tick (*A. americanum*), a species not found in Brazil. In this study, the deer were infested with *R. microplus* and *A. sculptum*. According to Calchi et al. [36], multiple tick species may play a role in *Theileria* spp. transmission among deer from Brazil. This study reinforces the hypothesis that undescribed *Theileria* species are circulating in deer populations within the country [36].

All tapirs sampled in this study tested positive for piroplasmids, a prevalence higher than that reported in previous studies in Brazil. For example, Mongruel et al. [11] recorded a prevalence of 52.5% in tapirs from the Brazilian Cerrado and Pantanal. Phylogenetic analyses based on the near-complete 18S rRNA and *hsp70* genes revealed that the *Theileria* sequences obtained from tapirs clustered with *Theileria terrestris* previously described in lowland tapirs in the Cerrado and Pantanal biomes in Mato Grosso do Sul state, Brazil [11]. All sampled tapirs carried ticks, including *A. sculptum*, *Amblyomma* spp. larvae, and *R. microplus*. Considering the well-known intense parasitism of *A. sculptum* on tapirs in the Cerrado [49] and the high parasitism rate observed in this study, this tick species could be a potential vector. However, further studies are necessary to characterize its biological role, identify additional vertebrate hosts, and clarify the transmission cycles involved.

To the best of the authors’ knowledge, this study represents the first report of *C. brasiliensis* in a cougar in Cerrado biome, further supporting its classification as a unique taxon within the *Cytauxzoon* genus, distinct from *C. felis* [48]. This finding challenges previous reports attributing *C. felis* as the only *Cytauxzoon* species infecting South American felids. Indeed, the supposed high genetic similarity with the pathogenic *C. felis* was based on short 18S rRNA gene sequences in most studies previously performed in Brazil, which may not provide sufficient resolution for accurate taxonomic classification. Our results reinforce previous detections of *Cytauxzoon* spp. in *P. concolor* from Goiás, Brazil [2], and suggest a broader distribution of this newly described piroplasmid in South American felids, distinct from *C. felis* infecting felids in North America [48]. Nonetheless, Calchi et al. [50] reported the occurrence of several *C. felis* genotypes in jaguars (*Panthera onca*) from distinct ecotypes from Brazil. Until now, *C. brasiliensis* has been detected in little-spotted cats (*Leopardus tigrinus*) [48], ocelots (*Leopardus pardalis*), and domestic cats from Brazil [50].

Furthermore, our findings reinforce the hypothesis that cougars may serve as primary hosts for *Cytauxzoon* spp. in South America, given their wide distribution, abundance, and high infection rates [2,50,51]. In this study, 83.3% (5/6) of *P. concolor* individuals tested positive for *Cytauxzoon* spp. by PCR. Additionally, three infected individuals were parasitized by *Amblyomma* spp. larvae. These findings highlight the need for further studies to identify potential tick vectors and clarify the transmission dynamics of *C. brasiliensis* in wild felid populations. Previous molecular and epidemiological studies incriminated *A. sculptum* as a possible vector of *Cytauxzoon* spp. in Brazil [52].

This study also detected *B. goianiaensis* in capybaras, aligning with previous records in Goiás, suggesting the possible endemicity of this parasite in capybara populations in this state [12]. Concerning the findings of a short 18S rRNA sequence showing high identity to *B. vogeli* in *C. prehensilis*, the absence of a longer sequence precluded inferring a definitive taxonomic positioning of this piroplasmid. Nonetheless, the finding suggests that the host range of *B. vogeli* or a closely related genotype may be broader than previously assumed. A similar result was reported by Sousa et al. [53], who identified a *B. vogeli*-related 18S rRNA genotype in rodents from the Pantanal, Brazil, reinforcing the potential diversity and adaptability of these hemoparasites among Neotropical mammals. While co-feeding transmission by tick vectors cannot be entirely ruled out, the *C. prehensilis* specimen analyzed in this study was sampled in a wildlife rehabilitation center in Goiânia. Domestic dogs, the primary hosts for *B. vogeli*, are present in this urban area [4,7]. Future studies are needed to amplify the near-complete 18S rRNA and additional molecular markers to clarify this *Babesia* genotype’s identity and phylogenetic relationships.

The presence of piroplasmids in wild mammals raises concerns regarding potential transmission to domestic animals and humans. The increasing fragmentation of habitats and expansion of urban areas heighten the risk of pathogen spillover, reinforcing the importance of a One Health approach to disease surveillance. Since some piroplasmids are zoonotic, further investigation is warranted to determine their pathogenic potential in different hosts [4,10,38].

From a conservation perspective, piroplasmid infections could represent an additional stressor for endangered wildlife species, particularly those under rehabilitation or captivity. Surveillance programs targeting piroplasmids and associated vectors are essential for contributing to conservation strategies and mitigating health risks for wildlife and domestic species.

Despite the valuable insights provided by this study, certain limitations must be acknowledged. The sample size, though comprehensive, was constrained by logistical challenges in capturing and sampling wild mammals. Additionally, short-fragment sequencing restricted the ability to characterize some piroplasmid genotypes fully. Future research should focus on whole-genome sequencing approaches to resolve taxonomic uncertainties and identify potential new species. Longitudinal studies on tick-host interactions and vector competence are necessary to clarify transmission dynamics.

## 5. Conclusions

This study expands current knowledge on the diversity and distribution of piroplasmids in Brazilian wild mammals, reporting the occurrence of *C. brasiliensis* in *P. concolor*, *T. terrestris* in *Tapirus terrestris* and *Theileria* in *S. gouazoubira*. These findings highlight the importance of continued surveillance of tick-borne pathogens, which may negatively impact One Health initiatives, biodiversity conservation, and the efforts to prevent emerging diseases. Expanding molecular surveillance across different biomes and host species will be essential to unravel the transmission dynamics of piroplasmids in South America and beyond.

## Figures and Tables

**Figure 1 pathogens-14-00585-f001:**
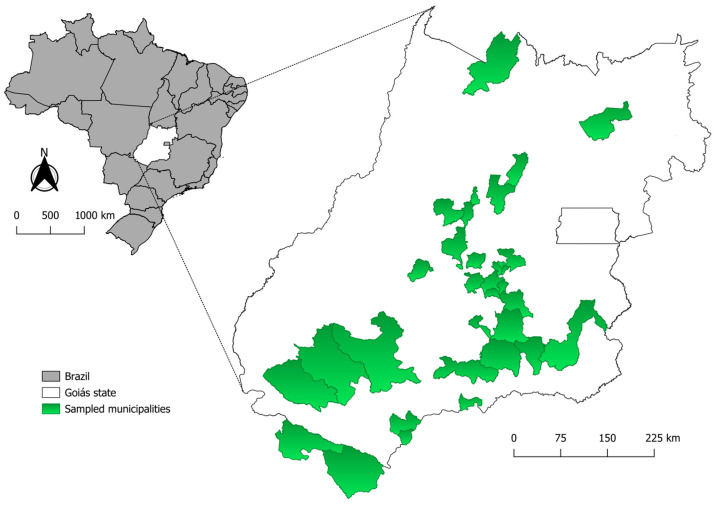
Map illustrating the distribution of animals sampled, considering the capture location, in Goiás state, midwestern Brazil.

**Figure 2 pathogens-14-00585-f002:**
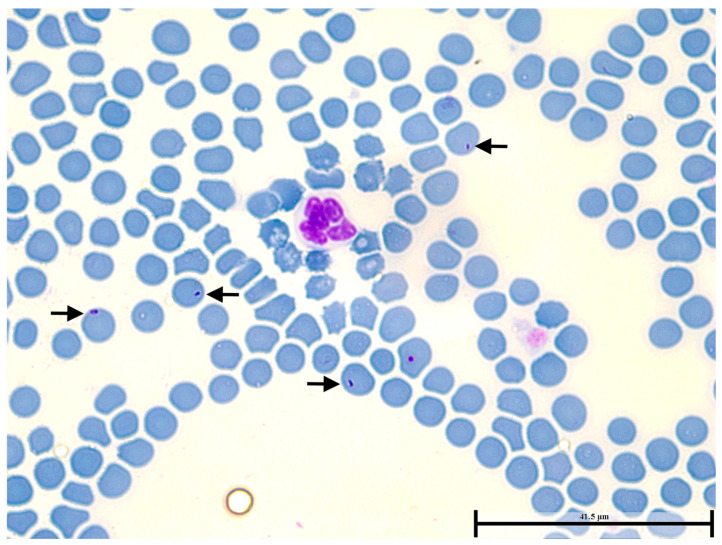
Intraerythrocytic inclusions (arrowed) compatible with *Cytauxzoon* spp. observed at oil immersion (1000× magnification) in stained-blood smears of a *P. concolor* captured in the municipality of Goiânia, Goiás state, midwestern Brazil.

**Figure 3 pathogens-14-00585-f003:**
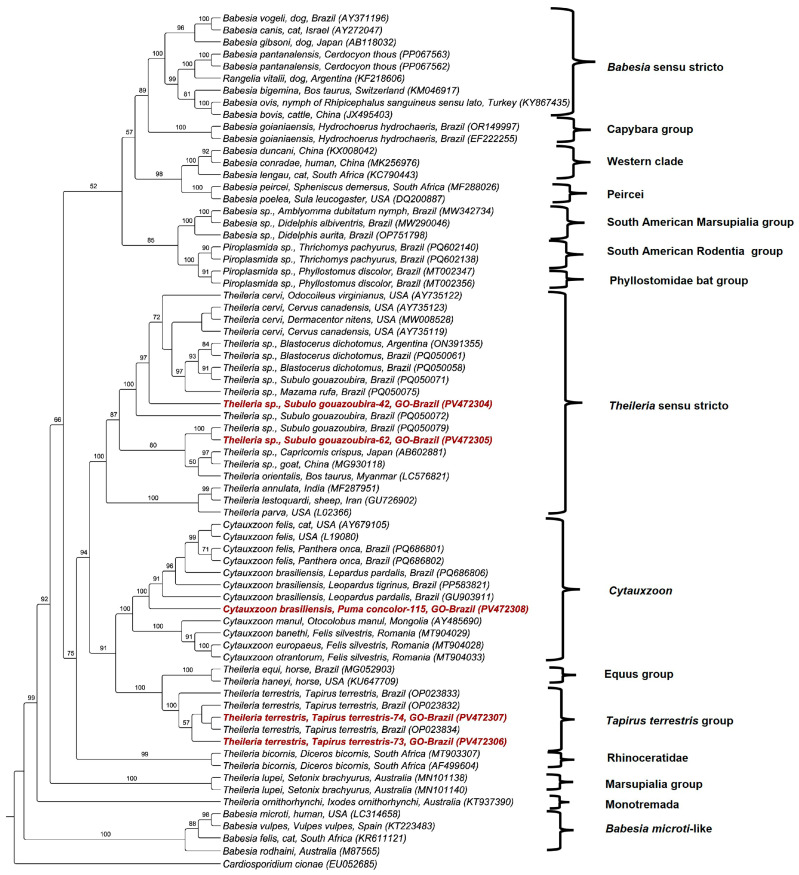
Phylogenetic analysis of Piroplasmid 18S rRNA sequences inferred from a 1610 bp alignment generated using the maximum likelihood method and the TIM3+I+G evolutionary model. Sequences obtained in the present study were highlighted in red. *Cardiosporidium cionae* was used as an outgroup.

**Figure 4 pathogens-14-00585-f004:**
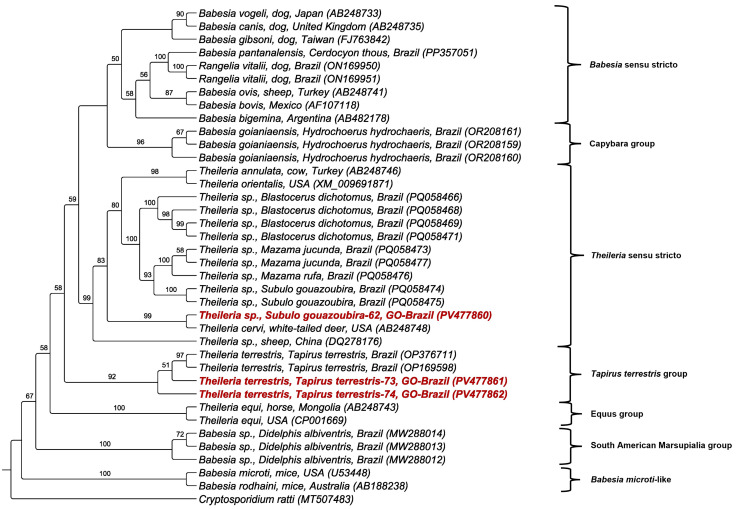
Phylogenetic analysis of piroplasmid *hsp70* sequences inferred from a 909 bp alignment generated using the maximum likelihood method and TIM3e+I+G evolutionary model. Sequences obtained in the present study were highlighted in red. *Cryptosporidium ratti* was used as an outgroup.

**Figure 5 pathogens-14-00585-f005:**
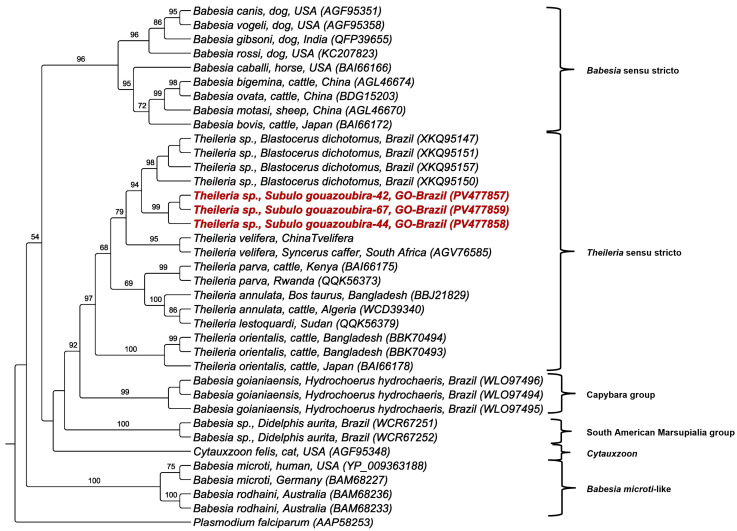
Phylogenetic tree inferred from the alignment of 242 amino acids of the *cox3* gene, using the maximum likelihood method and mtZOA+G evolutionary model. Sequences obtained in the present study were highlighted in red. *Theileria* spp. sequences detected in *S. gouazoubira* from Brazil formed a well-supported monophyletic clade, close to the *Theileria* sp. sequences obtained from *B. dichotomus* from Brazil. Support values (bootstrap) are indicated at the nodes. *Plasmodium falciparum* was used as an outgroup.

**Figure 6 pathogens-14-00585-f006:**
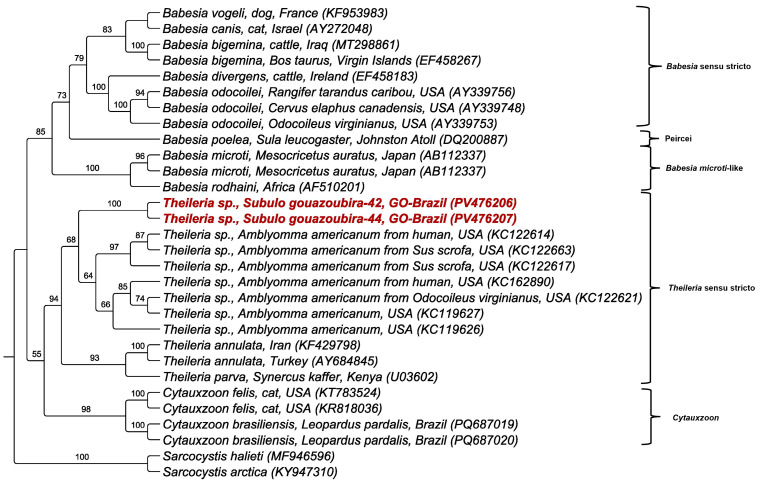
Phylogenetic tree based on the analysis of the ITS-1 intergenic region (2851 characters) using the maximum likelihood method and HKY+I+G evolutionary model. The values at the nodes indicate the bootstrap supports. *Theileria* spp. sequences detected in *S. gouazoubira* in Brazil are highlighted in red. *Sarcocystis halieti* and *Sarcocystis arctica* were used as an outgroup.

**Table 1 pathogens-14-00585-t001:** Collection sites and number of mammals from each species sampled in Goiás state, midwestern Brazil, from April 2023 to January 2024.

Mammal Species	CETAS-GO	CETRAS-CN	SMA	Vet Clinics	ZOO GYN	Total Number of Animals
*Alouatta caraia*	11	-	-	-	-	11
*Callithrix penicillata*	-	1	-	-	-	1
*Cavia porcellus*	-	-	-	1	-	1
*Cerdocyon thous*	11	-	-	-	-	11
*Chrysocyon brachyurus*	6	-	-	-	-	6
*Coendou prehensilis*	5	1	-	-	-	6
*Didelphys albiventris*	3	1	-	1	-	5
*Euphractus sexcinctus*	1	-	-	-	-	1
*Hydrochoerus hydrochaeris*	3	-	1	-	-	4
*Leopardus pardalis*	1	-	-	-	-	1
*Mazama americana*	1	-	-	-	-	1
*Myrmecophaga tridactyla*	21	-	-	-	-	21
*Nasua nasua*	-	-	-	-	1	1
*Oryctolagus cuniculus*	-	-	1	4	-	5
*Pecari tajacu*	-	1	-	-	-	1
*Puma concolor*	6	-	-	-	-	6
*Rattus norvegicus f. domestica*	-	-	-	-	2	2
*Sapajus libidinosus*	4	-	-	-	-	4
*Subulo gouazoubira*	11	1	-	-	-	12
*Tamandua tetradactyla*	1	-	-	-	-	1
*Tapirus terrestris*	2	-	-	-	2	4
Total	87	5	2	6	5	105

CETAS-GO: Wild Animal Screening Center of Goiás; CETRAS-CN: Screening and Rehabilitation Center for Wild Animals of Caldas Novas; SMA: Environmental Secretariat of the municipality of Anápolis; Vet Clinics: Veterinary Clinics; ZOO GYN: Goiânia Zoo, -: 0 samples collected.

**Table 2 pathogens-14-00585-t002:** Molecular detection of piroplasmid DNA in ticks collected from wild mammals and unconventional pets sampled in Goiás state, midwestern Brazil, from April 2023 to January 2024.

Tick Species	Tick Stage (No. of Specimens)	Host Species (No. of Animals)	Samples Tested by PCR *	Piroplasmid DNA-Positive/No. Tested (% of Positives)
*Amblyomma sculptum*	Nymph (123)	*A. caraya* (1), *C. prehensilis* (1), *C. thous* (1), *H. hydrochaeris* (2), *S. gouazoubira* (1), *M. tridactyla* (5), *T. terrestris* (2), *T. tetradactyla* (3)	37	2/37 (5.4%)
	Males (40)	*M. tridactyla* (8), *T. terrestris* (2)	15	0/15 (0%)
	Females (33)	*H. hydrochaeris* (1), *M. tridactyla* (4); *T. terrestris* (2)	7	0/7 (0%)
*Amblyomma dubitatum*	Males (32)	*H. hydrochaeris* (2)	5	1/5 (20%)
	Females (6)	*H. hydrochaeris* (2)	0	-
*Rhipicephalus microplus*	Females (3)	*S. gouazoubira* (1), *T. terrestris* (1)	0	-
*Amblyomma nodosum*	Males (17)	*M. tridactyla* (4), *T. tetradactyla* (4)	5	0/5 (0%)
	Females (8)	*M. tridactyla* (5), *T. tetradactyla* (2)	0	-
*Dermacentor nitens*	Males (2)	*P. concolor* (1)	0	-
*Amblyomma longirostre*	Females (1)	*C. prehensilis* (1)	0	-
	Males (3)	*C. prehensilis* (1)	0	-
*Amblyomma ovale*	Males (2)	*P. concolor* (1)	0	-
	Females (2)	*P. concolor* (1)	0	-
*Amblyomma calcaratum*	Males (4)	*M. tridactyla* (1), *T. tetradactyla* (1)	0	-
*Amblyomma* spp.	Larvae (24)	*M. tridactyla* (1), *P. concolor* (3), *T. terrestris* (1)	0	-
Total	300		69	3/69 (4.3%)

No: number; *: number of samples that successfully underwent DNA extraction.

**Table 3 pathogens-14-00585-t003:** Number and species of animals positive for piroplasmid DNA in Goiás state, midwestern Brazil, from April 2023 to January 2024.

Mammal Species	No. per Species (% of Mammals Collected)	Piroplasmid DNA-Positive/No. of Tested (% of Positives)
*Alouatta caraia*	11 (10.5%)	0/11 (0%)
*Callithrix penicillata*	1 (0.9%)	0/1 (0%)
*Cavia porcellus*	1 (0.9%)	0/1 (0%)
*Cerdocyon thous*	11 (10.5%)	2/11 (18.2%)
*Chrysocyon brachyurus*	6 (5.7%)	2/6 (33.4%)
*Coendou prehensilis*	6 (5.7%)	1/6 (16.7%)
*Didelphys albiventris*	5 (4.8%)	0/5 (0%)
*Euphractus sexcinctus*	1 (0.9%)	0/1 (0%)
*Hydrochoerus hydrochaeris*	4 (3.9%)	2/4 (50%)
*Leopardus pardalis*	1 (0.9%)	0/1 (0%)
*Mazama americana*	1 (0.9%)	1/1 (100%)
*Myrmecophaga tridactyla*	21 (20%)	2/21 (9.5%)
*Nasua nasua*	1 (0.9%)	0/1 (0%)
*Oryctolagus cuniculus*	5 (4.8%)	0/5 (0%)
*Pecari tajacu*	1 (0.9%)	0/1 (0%)
*Puma concolor*	6 (5.7%)	4/6 (66.7%)
*Rattus norvegicus*	2 (1.9%)	0/2 (0%)
*Sapajus libidinosus*	4 (3.9%)	0/4 (0%)
*Subulo gouazoubira*	12 (11.5%)	9/12 (75%)
*Tamandua tetradactyla*	1 (0.9%)	0/1 (0%)
*Tapirus terrestris*	4 (3.9%)	4/4 (100%)
Total	105 (100%)	27/105 (25.7%)

No: number.

**Table 4 pathogens-14-00585-t004:** BLASTn results of the sequences of four distinct molecular markers for piroplasmids detected in mammals from the midwestern region of Brazil.

Sample No. (GenBank Accession No.) Wild Mammal Species	Molecular Marker	Size (bp)	Percent Identity	Query Cover (%)/E-Value	Sequence Best Match Location/Host (GenBank Accession No.)
Sample 42 (PV472304/PV477857/PV476206) *Subulo gouazoubira*	18S rRNA	1352	99.93%	100%/0.0	*Theileria cervi* Wisconsin—USA/elk blood (AY735123.1)
	*cox-3*	650	87.50%	100%/0.0	*Theileria* sp. Brazil/*Blastocerus dichotomus* (PQ058478.1)
	ITS-1	566	89.60%	100%/0.0	*Theileria* sp. Tennessee—USA/*Amblyomma americanum* from *Odocoileus virginianus* (KC122621.1)
Sample 44 (PV477858/PV476207) *Subulo gouazoubira*	*cox-3*	650	87.62%	99%/0.0	*Theileria* sp. Brazil/*Blastocerus dichotomus* (PQ058488.1)
	ITS-1	534	89.17%	100%/0.0	*Theileria* sp. Georgia—USA/*Amblyomma americanum* (KC119626.1)
Sample 45 (PV544933) *Hydrochoerus hydrochaeris*	18S rRNA	565	100%	100%/0.0	*Babesia goianiaensis* Goiás—Brazil/*Hydrochoerus hydrochaeris* (OR149999.1)
Sample 62 (PV472305/PV477860) *Subulo gouazoubira*	18S rRNA	1358	99.41%	100%/0.0	*Theileria cervi* Texas—EUA/*Dermacentor nitens* from *Odocoileus virginianus* (MW008521.1)
	*hsp70*	810	92.92%	100%/0.0	*Theileria cervi* Georgia—USA/*Odocoileus virginianus* (AB248748.1)
Sample 67 (PV477859) *Subulo gouazoubira*	*cox-3*	650	87.62%	99%/0.0	*Theileria* sp. Brazil/*Blastocerus dichotomus* (PQ058488.1)
Sample 73 (PV472306/PV477861) *Tapirus terrestris*	18S rRNA	1350	100%	100%/0.0	*Theileria* sp. Brazil/*Tapirus terrestris* (OP023832.1)
	*hsp70*	893	99.22%	86%/0.0	*Theileria* sp. Brazil/*Tapirus terrestris* (OP376711.1)
Sample 74 (PV472307/PV477862) *Tapirus terrestris*	18S rRNA	1397	100%	100%/0.0	*Theileria* sp. Brazil/*Tapirus terrestris* (OP023832.1)
	*hsp70*	739	99.04%	99%/0.0	*Theileria* sp. Brazil/*Tapirus terrestris* (OP169598.1)
Sample 108 PV544934 *Subulo gouazoubira*	18S rRNA	437	100%	100%/0.0	*Theileria cervi* Indiana—EUA/elk blood (AY735119.1)
Sample 115 (PV472308) *Puma concolor*	18S rRNA	1270	99.76%	100%/0.0	*Cytauxzoon brasiliensis* Brazil/*Leopardus pardalis* (PP583821.1)
Sample 128 (PV544935) *Coendou prehensilis*	18S rRNA	304	100%	100%/1e-155	*Babesia vogeli* Brazil/*Canis lupus familiaris* (MN912657.1)

## Data Availability

The data presented in this study are available within this article.

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
