# Peer review of "Survey of Piroplasmids in Wild Mammals, Unconventional Pets, and Ticks from Goiás State, Midwestern Brazil"

_pathogens, 2025, doi:10.3390/pathogens14060585_

Round 1
Reviewer 1 Report
Comments and Suggestions for Authors
In this study the authors examine the presence and diversity of piroplasms in wildlife and non-traditional mammalian pets using samples from wildlife rescue centers and veterinary clinics in Goiás State, Brazil. The research uses well known and accepted methodology, and the document is well written. This study expands current knowledge on the diversity and distribution of piroplasmids in Brazilian wild mammals.
Minor comments:
-Line 38. Delete the word clustered
-Line 199. Include the other genes (size of the alignment and evolutionary method).
-Line 237. What about the results from the cytB amplifications that you mentioned on line 158?
-Line 264. Only four samples out of the 27 blood + 3 ticks where successfully sequenced? why was that? the sequencing rate for all the other markers is also very low, only three CO3 sequences, etc. Did you try to sequence all the positives? Or just a few? Why? Include in the discussion.
-Were there any clinical signs on the animals that were sampled and positive? Discuss either way
Author Response
Reviewer 2:
In this study the authors examine the presence and diversity of piroplasms in
wildlife and non-traditional mammalian pets using samples from wildlife rescue
centers and veterinary clinics in Goiás State, Brazil. The research uses well
known and accepted methodology, and the document is well written. This study
expands current knowledge on the diversity and distribution of piroplasmids in
Brazilian wild mammals.
Response:
Thank you very much for your kind and encouraging comments. We sincerely
appreciate your positive feedback on our study. All your suggestions and
observations have been carefully addressed and incorporated into the revised
manuscript.
Reviewer 2: -Line 38. Delete the word clustered
Response: Thanks. We deleted.
Reviewer 2: -Line 199. Include the other genes (size of the alignment and
evolutionary method).
Response:
We thank the reviewer for this valuable observation. We have now included the
information regarding the alignment sizes and evolutionary models used for each
gene in the phylogenetic analyses, as suggested.
Reviewer 2:
-Line 237. What about the results from the cytB amplifications that
you mentioned on line 158?
Response:
Thank you for the important suggestion. We have added the information to the
manuscript.
Reviewer 2: -Line 264. Only four samples out of the 27 blood + 3 ticks where
successfully sequenced? why was that? the sequencing rate for all the other
markers is also very low, only three CO3 sequences, etc. Did you try to sequence
all the positives? Or just a few? Why? Include in the discussion.
Response:
Thank you for this important observation. Indeed, all positive samples were
initially selected for sequencing. However, sequencing success was limited due
to low DNA concentration and/or poor amplicon quality in several cases,
particularly for longer or more variable gene regions. These technical limitations
have now been clarified and discussed in the revised manuscript, as suggested.
Reviewer 2: -Were there any clinical signs on the animals that were sampled and
positive? Discuss either way
Response:
Thank you for your observation. None of the sampled animals that tested positive
for piroplasmids exhibited specific clinical signs attributable to infection at the time
of sampling. This information has now been briefly addressed in the discussion,
as suggested.
Reviewer 2 Report
Comments and Suggestions for Authors
Dear Editor and authors,
Here are my comments for: Survey of Piroplasmids in Wild Mammals, Unconventional Pets, 2 and Ticks from Goiás State, Midwestern Brazil.
General
This paper describes a survey of piroplasm in central Brazil, based on incidentally acquired material. The opportunistic sampling led to uneven numbers among hosts, and locations, presumably due to varying sampling effort. Ticks, blood, and tissue were analyzed, by state-of-the-art methodologies, and the findings were reported and discussed.
Opportunistic sampling rarely leads to highly significant findings due to the quality of the data. This is also the case for this study which due to modest sample size and uneven distribution, has limitations. While I do not find the outcome very informative, I also acknowledge that this type of survey provides baseline data on the current state, which may be critical as a reference for later studies. In contrast, I find the methodology and quality of the writing very high, and it was quite pleasing to note that the authors have interpreted their findings, with the level of caution that I would recommend.
The grammar and language are nearly flawless, and the paper is easily read. I am left with the impression that they did write the first draft themselves, but I suspect that the authors have improved the manuscript by use of AI, for readability. The editor should check the submitted information on the use of AI.
The print scale of the phylogenetic trees is, however, too small and must be increased in size. Pls reconsider the use of bold text in the tables, just as you could reconsider the width of the individual columns – for readability, e.g. Merge lines in table 2 such that genus and species names are divided over two lines and let each host species name appear in one line.
Major
None
Minor.
Line 300. Add “in” after detected.
Line 444. Phylogenetic rather than evolutionary.
Closing remark.
I am aware that the above might seem as if I uncritically accepted the paper, especially since I normally return 2 to 3 pages with review comments. However, several reads did not produce more than noted above.
Author Response
Reviewer 1:
Here are my comments for: Survey of Piroplasmids in Wild Mammals,
Unconventional Pets, 2 and Ticks from Goiás State, Midwestern Brazil.
General:
This paper describes a survey of piroplasm in central Brazil, based on incidentally
acquired material. The opportunistic sampling led to uneven numbers among
hosts, and locations, presumably due to varying sampling effort. Ticks, blood, and
tissue were analyzed, by state-of-the-art methodologies, and the findings were
reported and discussed.
Opportunistic sampling rarely leads to highly significant findings due to the quality
of the data. This is also the case for this study which due to modest sample size
and uneven distribution, has limitations. While I do not find the outcome very
informative, I also acknowledge that this type of survey provides baseline data
on the current state, which may be critical as a reference for later studies. In
contrast, I find the methodology and quality of the writing very high, and it was
quite pleasing to note that the authors have interpreted their findings, with the
level of caution that I would recommend.
The grammar and language are nearly flawless, and the paper is easily read. I
am left with the impression that they did write the first draft themselves, but I
suspect that the authors have improved the manuscript by use of AI, for
readability. The editor should check the submitted information on the use of AI.
The print scale of the phylogenetic trees is, however, too small and must be
increased in size. Pls reconsider the use of bold text in the tables, just as you
could reconsider the width of the individual columns – for readability, e.g. Merge
lines in table 2 such that genus and species names are divided over two lines
and let each host species name appear in one line.
Response:
Thank you very much for your kind comments and thoughtful suggestions. We
truly appreciate your positive feedback on our methodology and writing. All your
recommendations, including the adjustments to the phylogenetic trees and
tables, have been carefully addressed in the revised manuscript.
We appreciate your suggestion regarding the table formatting. However, it was
not possible to adjust the column widths without compromising the overall layout,
given the extensive and valuable data included. Nevertheless, we have followed
your recommendation regarding text emphasis and have retained bold formatting
only for the column headers.
Reviewer 1: Line 300. Add “in” after detected.
Response: We added. Thanks
Reviewer 1: Line 444. Phylogenetic rather than evolutionary.
Response: Thanks. We changed.
Reviewer 1: Closing remark.
I am aware that the above might seem as if I uncritically accepted the paper,
especially since I normally return 2 to 3 pages with review comments. However,
several reads did not produce more than noted above.
Response:
Thank you very much for your compliments and valuable suggestions.
Reviewer 3 Report
Comments and Suggestions for Authors
The manuscript is very well written, and the analysis is of a high standard. The sample size is large enough for the time interval examined to be relatively short. In terms of content, the article meets all the criteria for publication in a scientific journal. However, there are several technical errors in the text:
1. The number of samples/animals in this form (n = 5) is inconsistent. In some places, 'n' is written in italics; in others, it is not. In some places, there are spaces in brackets; in others, there are none. This needs to be standardised.
2. Lines 226–233: Some small numbers should be written in words, as in other parts of the manuscript.
Author Response
Reviewer 3:
The manuscript is very well written, and the analysis is of a high standard. The
sample size is large enough for the time interval examined to be relatively short.
In terms of content, the article meets all the criteria for publication in a scientific
journal. However, there are several technical errors in the text:
Response:
Thank you very much for your kind comments and positive evaluation of our work.
We also appreciate your attention to detail and have carefully corrected all the
technical issues noted in the manuscript.
Reviewer 3:
1. The number of samples/animals in this form (n = 5) is inconsistent. In some
places, 'n' is written in italics; in others, it is not. In some places, there are spaces
in brackets; in others, there are none. This needs to be standardised.
Response:
Thank you for your valuable observation. All inconsistencies related to the
formatting of sample numbers (e.g., n = 5) have been carefully reviewed and
corrected throughout the manuscript to ensure standardization.
Reviewer 3:
2. Lines 226–233: Some small numbers should be written in words, as in other
parts of the manuscript.
Response:
Thank you very much for the important correction. We have made
the necessary changes.